# Seamless Fusion: Multi-Modal Localization for First Responders in Challenging Environments

**DOI:** 10.3390/s24092864

**Published:** 2024-04-30

**Authors:** Dennis Dahlke, Petros Drakoulis, Anaida Fernández García, Susanna Kaiser, Sotiris Karavarsamis, Michail Mallis, William Oliff, Georgia Sakellari, Alberto Belmonte-Hernández, Federico Alvarez, Dimitrios Zarpalas

**Affiliations:** 1German Aerospace Center (DLR), 12489 Berlin, Germany; dennis.dahlke@dlr.de; 2Visual Computing Lab, Information Technologies Institute, Centre for Research and Technology Hellas (CERTH), 57001 Thermi, Greece; petros.drakoulis@iti.gr (P.D.); skaravarsamis@iti.gr (S.K.); michmall@iti.gr (M.M.); zarpalas@iti.gr (D.Z.); 3Señales, Sistemas y Radiocomunicaciones, ETSI Telecomunicación, Universidad Politécnica de Madrid (UPM), 28040 Madrid, Spain; anaida.fernandez@upm.es (A.F.G.); abh@gatv.ssr.upm.es (A.B.-H.); 4German Aerospace Center (DLR), 82234 Wessling, Germany; susanna.kaiser@dlr.de; 5CS2 Research Centre, School of Computing and Mathematical Sciences, University of Greenwich, London SE10 9LS, UK; william.oliff@greenwich.ac.uk (W.O.); g.sakellari@greenwich.ac.uk (G.S.)

**Keywords:** multi-modal localization, self-localization, seamless fusion, sensor fusion, first responders, visual localization, Galileo satellite navigation, inertial navigation

## Abstract

In dynamic and unpredictable environments, the precise localization of first responders and rescuers is crucial for effective incident response. This paper introduces a novel approach leveraging three complementary localization modalities: visual-based, Galileo-based, and inertial-based. Each modality contributes uniquely to the final Fusion tool, facilitating seamless indoor and outdoor localization, offering a robust and accurate localization solution without reliance on pre-existing infrastructure, essential for maintaining responder safety and optimizing operational effectiveness. The visual-based localization method utilizes an RGB camera coupled with a modified implementation of the ORB-SLAM2 method, enabling operation with or without prior area scanning. The Galileo-based localization method employs a lightweight prototype equipped with a high-accuracy GNSS receiver board, tailored to meet the specific needs of first responders. The inertial-based localization method utilizes sensor fusion, primarily leveraging smartphone inertial measurement units, to predict and adjust first responders’ positions incrementally, compensating for the GPS signal attenuation indoors. A comprehensive validation test involving various environmental conditions was carried out to demonstrate the efficacy of the proposed fused localization tool. Our results show that our proposed solution always provides a location regardless of the conditions (indoors, outdoors, etc.), with an overall mean error of 1.73 m.

## 1. Introduction

Thousands of rescue operations (ROs) are conducted each year all over the world. Detailed US statistics [1], indicative of the developed world, reveal that the lives saved/lives lost ratio, after first responder (FR) intervention, progressed from 14.9 in the 1990s to 22.6 in the 2000s and to 23 in the 2010s. While the empirical data pinpoint that the operational efficiency is reaching a plateau, globalization-induced mobility and climate change-related factors, together with the global population increase, continue to fuel the number of incidents that require FR attention. ROs are inherently stressful events where minor delays in decisionmaking or suboptimal procedures, such as inaccurate pathfinding or unwise task assignments due to incomplete or incorrect field perception, can easily become fatal. This not only poses risks to the victims but also endangers the operators involved. The International Forum to Advance First Responder Innovation has identified ten common global capability gaps, with “the ability to know the location of responders and their proximity to risks and hazards in real time” [2] ranking as the number one gap. This underscores the critical importance of self-localization and accurate positioning as errors in the provided locations can lead to disorientation and potentially create new dangerous situations. Therefore, reliable and accurate positioning is a fundamental requirement for any self-localization technique.

Location-based systems have proven to be essential tools not only in emergency operations, leveraging a wide array of technologies such as WiFi, Bluetooth, other wireless sensors, cameras, the GPS, GNSS devices, and drones, but also in a multitude of other critical application areas. Beyond emergency response, these technologies have been significantly applied in smart energy management, intelligent HVAC (Heating, Ventilation, and Air Conditioning) controls, point-of-interest identification, and occupancy prediction. For instance, intelligent energy management systems utilize the location to optimize the energy consumption based on actual and forecasted building occupancy [3]. Similarly, in the realm of HVAC control, the real-time location allows for the dynamic adjustment of the temperature and air quality in different zones, thus enhancing the efficiency and comfort [4]. In the context of identifying points of interest and predicting occupancy, localization technologies enable systems to provide users with relevant contextual information, as well as to anticipate space usage patterns for more efficient management [5]. These applications demonstrate the versatility and value of location-based systems in improving the operational efficiency, safety, and user experience across multiple domains.

While wireless signal-based techniques, such as WiFi, Bluetooth, and UWB, along with fingerprinting methods, have demonstrated significant potential in location-based services, their application in emergency and rescue operations faces inherent limitations [6]. These techniques generally require pre-deployed infrastructure or nodes, which can be a major constraint in disaster scenarios where the existing infrastructure may be damaged or entirely non-functional [7]. Furthermore, fingerprinting methods necessitate the prior knowledge of the environment through the collection of signal strength maps, a process that is not feasible in the dynamic or rapidly changing conditions typical of emergency situations [8]. The reliability of these techniques can also be severely compromised by environmental factors, such as interference, obstructions, and signal attenuation, which are often exacerbated in chaotic environments [9]. In addition, the deployment and maintenance of the required infrastructure can be costly and time-consuming, limiting the scalability and flexibility needed for an effective emergency response. Given these challenges, relying solely on these signal-based and fingerprinting techniques for emergency and rescue tasks may not provide the robustness and immediacy needed to operate effectively in critical life-threatening situations.

In addressing these common in-field challenges, the aim of this work is to empower FRs and their commanding entities with the real-time spatial information they need to consistently achieve their objectives. Specifically, we propose and evaluate a multi-modal localization system tailored to the needs of FRs operating in dynamic and unpredictable environments. By seamlessly integrating data from inertial sensors, a Global Navigation Satellite System (GNSS), and visual sensors, our system aims to provide accurate and reliable positioning information both indoors and outdoors. Fusing information from diverse sources can mitigate the innate limitations of individual modalities and potentially provide more robust positioning information, particularly in challenging environments where a single modality may be insufficient. From an operational point of view, seamless tracking is essential for swift and correct decisionmaking during distressing situations. By providing FRs with visualized current locations within the area of interest, they can make better judgments about their movements. This heightened level of awareness is expected to improve various aspects of FRs’ operational performance, including better area coverage, spatially reliable reporting, improved pathfinding, and the avoidance of threats related to disorientation. Finally, from a commander’s point of view, the visualized real-time location information of all the team members greatly simplifies commanding and enhances the reactions to potential mishaps.

The novelty and primary aim of this paper can be summarized as follows:The FRs need an all-encompassing localization approach. It should provide reasonable locations in all environments, like indoor/outdoor scenarios, dark rooms, smoky/dusty environments, and under harsh conditions. We provide three different tools that are designed for different conditions and that are able to complement each other. Depending on the scenario and in the case that one tool is not working properly, the system is able to automatically switch to the results of another tool that is able to provide a location update in that respective environment. The incorporation of various situations and solutions marks a significant advancement over previous studies [10,11,12], which depend solely on a single method for a specific scenario. Additionally, the majority of the existing research relies on Wireless Sensor Networks that are pre-deployed in the monitored environments [13,14,15].Typically, all the available signals are fused in one algorithm—e.g., in a Simultaneous Localization and Mapping (SLAM) [16,17] approach or another sensor fusion approach [18,19,20]—to provide reasonable localization results. The complexity of sensor fusion escalates significantly when the sensors involved lack precision, often resulting in suboptimal localization and mapping outputs. Traditional methods that rely on integrating imprecise signals can lead to increased error propagation and reduced system reliability [21]. In contrast, our proposed high-level fusion method enhances the system’s accuracy and dependability. By leveraging the modularity and integrating multiple robust sources for positional data, our approach not only mitigates the issues associated with the precision of individual sensors but also ensures that the overall system remains resilient against sensor inaccuracies. In contrast to that, in this paper, we fuse the signals at a higher level. This approach provides modularity and robustness to the system since, at most times, there are multiple sources from which to retrieve positional information.In this combinatory but simultaneously redundant scheme, the individual tools can benefit from each other. For instance, the Visual self-localization tool can utilize the Fusion or Galileo tools to be initialized faster. Also, the inertial-based localization tool can utilize the rest of the tools to correct its expected drift so as to be as accurate as possible when it is really needed. Contrastingly, most of the previously presented works are contingent upon the simultaneous operation of all the methods and systems to perform effective fusion, thereby risking significant losses in accuracy if an additional sensor integral to the fusion process is absent or malfunctions. These approaches often lack the modular flexibility inherent in our system, which allows for independent operation or collaborative enhancement among the various tools.One of the pivotal strengths of this system lies in its development and testing within realistic operational environments, specifically tailored for use by FRs. Unlike the majority of the proposed solutions, which are often evaluated in controlled or simulated settings [22,23], this approach enables a direct comparison of the experimental tool results with real-world accuracy. Additionally, it assesses the system’s viability for real-time application by experienced FRs in active scenarios. This method not only underscores the practical relevance of our system but also enhances its reliability and effectiveness in genuine operational conditions.

This document is organized into five sections. The Introduction (Section 1) sets the groundwork for the study by defining the problem that is addressed and highlighting the main points of our solution. Next, the Related Work (Section 2) presents extensive background information on the individual localization modalities and on other fusion solutions. Subsequently, in the Methodology (Section 3), we present the details regarding our system architecture, the deployment platform, and all four of the tools developed. In this section, we describe the inner workings of each individual self-localization tool, as well as our proposed fused solution. Continuing, in the Results and Analysis (Section 4), we detail the experimental setup and present the outcomes of our study in both a quantitative and qualitative manner. Finally, in the Discussion (Section 5), we summarize the outcome of this work and provide some final remarks.

## 2. Related Work

In exploring the rich landscape of navigation and localization techniques, our investigation encompasses a nuanced examination of diverse methodologies. Our exploration incorporates different dimensions of localization, ranging from visual perception to the utilization of Galileo signals, the integration of inertial sensing technologies, as well as the fusion of the three. By considering each aspect independently in the following subsections, we gain a comprehensive understanding of their respective strengths and limitations.

### 2.1. Visual Self-Localization

The problem of self-localization is one of the tasks that, despite being natural to humans and most species—and we have clear evidence that it can be solved purely visually utilizing a combination of prior knowledge and intelligence—has been mostly approached indirectly through abstractions and modeling. Part of the reason is that Artificial Intelligence (AI) was not sophisticated enough to tackle the problem efficiently in the image domain, hence the indirect approach. This is why the most advanced solutions today are hybrid, also involving satellite and inertial positioning. Visual-based self-localization can be viewed as the new frontier of the field, where the upcoming rise of Artificial Intelligence meets the proliferation of miniaturized, powerful, and accessible devices, pushing mechanics a step forward towards the holy grail we call the “human condition”.

Visual localization has attracted much attention in recent years due to its key role in several tasks, such as virtual reality, augmented reality, robotics, autonomous driving, etc. [24]. There are two main approaches that dominate the field: structure-based methods and image-based methods. The former represent the scene via a 3D model and estimate the pose of a query image by directly matching 2D features to 3D points [25,26] or by matching 3D features to 3D points using a semantic representation of the scene [27]. On the other hand, image-based localization (IBL) was initially formulated as an image retrieval problem focused on matching a query image to an image database with geo-locations [28]. Following the recent advances in other computer vision tasks using deep learning, the authors in [29] introduced a Convolutional Neural Network (CNN) with a novel layer inspired by the Vector of Locally Aggregated Descriptors (VLAD). However, the features extracted from time-varying objects, such as pedestrians and trees, or ubiquitous objects, like vehicles and fences, can introduce misleading cues into the geo-localization process. Tackling these issues, the authors in [30] introduced an end-to-end Contextual Reweighting Network (CRN) that predicts the importance of each region in the feature map based on the image context. An alternative to image-based localization is direct 6DoF camera pose regression [31,32]. Extending PoseNet, the authors in [33] introduced more sophisticated loss functions, albeit information of the scene geometry is needed. Other approaches involve indexing-based techniques using a collection of previously collected panoramas [34]. Presently, the state of the art combines cross-view matching between images of distinct domains (aerial, panoramas, and perspective), and it can offer even higher accuracy orientation estimates [35]. Apart from treating the visual localization problem in its absolute form, which means to directly estimate the camera pose in a global or local frame of reference, in various cases, it can be equally useful and analogous to estimate the trajectory/path and position of the camera relative to an initial point of reference. Given the location of this point, the problem can be transformed into either a monocular VO [36] or a monocular SLAM [37,38], where new challenges arise, such as the estimation of the prediction scale and its correspondence to reality, which may require additional information or processing to be handled.

### 2.2. Galileo Self-Localization

Investigations on the use of Galileo signals alone [39], or in combination with the Global Positioning System (GPS) for navigation purposes, have been widely pursued since the beginning of Galileo [40] and the development and availability of Galileo receivers [41]. Since 2016, GNSS receivers have been available on almost every smartphone [42]. Additionally, a GNSS analysis tool accessing raw GNSS data is offered, for instance, by Google [43]. The raw GNSS data of the first smartphone enabling the reception of multi-frequency signals, namely the Xiaomi 8, were thoroughly investigated in [44] and compared to the data from high-precision devices. The positioning of the smartphone suffers mainly from duty cycling [45,46]. Because the phase is not continuously available, precise point positioning (PPP) is not always possible.

The work of [47] refers to the use of GNSS positioning in rescue-relevant scenarios, such as building assessment tasks. It is shown that, especially in difficult environments like under trees and in close proximity to man-made infrastructure, the time to obtain a first GNSS fix is considerably shorter when using a dedicated GNSS antenna. Furthermore, the use of dual-band receivers and antennas improves the positioning accuracy from worse than 3 m with the current smartphones to 1.4 m and better. Another aspect for using dedicated GNSS devices is the ability to adapt more easily to new technological improvements like the Galileo High-Accuracy Service (HAS), which has been available since January 2023 [48]. The HAS aims to provide a PPP service worldwide. It transmits precise orbits, clocks, and biases for both Galileo and the GPS, in the signal-in-space and through a ground channel [49]. This in-orbit service needs neither internet nor terrestrial infrastructure for the end user, and it provides a positioning accuracy of an order of magnitude better compared to the 1.5 m mentioned above [50]. One problem when dealing with broadcasting tracking information from FRs to incident commanders or to the command and control is low bandwidth or even connection loss. The possibility of buffering localization information in this case is investigated in [51].

### 2.3. Inertial Self-Localization

Being able to determine the position of people and objects reliably and accurately in indoor spaces has been a long-standing issue, with extensive research having been conducted in this area [52,53]. This is largely due to the wide range of applications and scenarios for which indoor positioning systems can be utilized. Indoor tracking/navigation approaches [54,55,56] are focused on the relative positioning information of a person from a starting location in an unknown area. These approaches do not depend on an offline survey process to be conducted beforehand and need to be able to operate in unknown environments, where an offline site survey process cannot be performed beforehand. Inertial self-localization approaches might require operation in infrastructureless environments [57], where there are no preinstalled devices, and to be able to obtain the principal data modalities without relying on any infrastructure of a building or indoor space, utilizing the onboard sensors of a smartphone device to obtain the positioning information of an individual. Relevant work has been conducted in [58], where the authors presented an Android-based smartphone to perform pedestrian dead reckoning (PDR) using inertial measurement unit (IMU) sensors, using a Kalman filter (KF) to perform the sensor fusion and derive a location estimate. However, aspects such as acceleration and velocity are not linear in nature [59], and, therefore, a standard KF is not appropriate as it assumes that all the inputs and outputs are Gaussian. In more recent literature, it has been more common to use the extended Kalman filter (EKF) and unscented Kalman filter (UKF) approaches, which are suitable for non-linear data, such as the work conducted in [60], which also used IMU sensors along with ultra-wide-band (UWB) technology to perform indoor position and navigation. Moreover, Particle Filtering (PF) [61] has also been used and shown to handle non-linear smartphone sensor data to provide high-accuracy indoor pedestrian positioning. On another note, IMU sensors are not always necessarily located on the user’s smartphone, as demonstrated in [62], where the authors used foot-mounted IMU sensors to conduct pedestrian positioning. However, these approaches require specific and more expensive equipment to deploy. On the other hand, PDR based on smartphone IMU sensors is a low-cost solution [56], but it presents a large challenge regarding drift PDR error [63], which occurs due to the errors that IMU sensors exhibit and accumulate over time. This accumulation error results in the difference between the actual location and the predicted location to increase over time. To help mitigate this issue, refs. [64,65,66] demonstrated that Received Signal Strength (RSS)-based techniques utilizing Bluetooth Low Energy (BLE) from the mobile device communications module can be leveraged to obtain the range estimation between itself and available landmark devices, such as a building black box.

### 2.4. Fusion Self-Localization

The concept of sensor fusion, in its most common form found in the literature, is usually about combining different modalities at a low level, usually under the umbrella of one multi-input procedure or tool. In our case, driven by the existence of the three autonomous self-localization tools and our goal to produce a redundant, robust, and modular system, we chose to engage with the combination of the individual modalities at a higher level.

The combined use of inertial navigation and Galileo signals was investigated in a project called SARHA (Sensor-Augmented Galileo Receiver for Handheld Applications in Urban and Indoor Environments) [67], assisted by a transponder system to be installed inside the building. The prototype demonstrates improved availability, reliability, and position localization performance in challenging environments, serving as an initial step toward integrating a GNSS with autonomous sensors for pedestrian navigation, although further integration and cost reduction are needed for mass-market applications. In the DingPos project [68], a system was introduced in which GPS and Galileo receivers were used in an indoor localization system assisted by map matching, ultra-wide-band (UWB) technology, and WiFi.

Currently, there are two projects funded by the European Union’s Horizon 2020 (H2020) research and innovation funding program aiming at the real-time tracking of FRs with different localization modalities. The H2020 PROTECT [69] features a system based on inertial sensors, called ARIANNA, specially designed for indoor tactical scenarios. Belt- or foot-mounted IMU sensors were augmented with a compass and an altimeter. Furthermore, indoor positioning accuracy is improved whenever floor plans are available. The H2020 INTREPID project [70] features the concept of a real-time positioning module (RPTM). The authors highlighted the strengths and weaknesses of individual sensors such as IMU, UWB, and SLAM based on stereo thermal cameras. The fusion of these sensors shows promise in reducing errors, but challenges such as system initialization and data synchronization remain. In addition to the methodologies discussed above, along with our own proposed approach, we would like to draw attention to the work in [71]. Their work introduced an Integrated Positioning System (IPS), which combines visual-aided inertial and standalone inertial navigation techniques, bolstered by reference localization sources such as the GPS. The IPS framework serves as a valuable indoor reference to validate our approach in the subsequent results section.

Examples of modular fusion localization systems are prevalent in the intelligent automotive domain. Moreover, the methods targeting the localization of pedestrians are more related to a use case that involves the localization of FRs or, in general, humans in motion. In the former domain, Wan et al. [72] estimated the optimal position, velocity, and attitude of the vehicle. The proposed system fuses information from GNSS, LiDAR, and IMU sensors to provide centimeter-precision localization accuracy by means of an error-state Kalman filter. A merit of the proposed system is that it exhibits operational resilience under challenging scenes. In a similar vein, Chen et al. [73] proposed a modular localization system that combines GNSS, LiDAR, and IMU subsystems. Their outputs were fused by a constraint Kalman filter. Wen et al. [74] used a fish-eye camera to capture the boundaries of the sky view in order to correct the non-line-of-sight (NLOS) and line-of-sight (LOS) aspects for the GNSS receiver. Then, a probabilistic factor graph was used to combine the data from the inertial sensor module (employing a gyroscope, a magnetometer, and an accelerometer) that provides a heading hint and the localization data from a GNSS receiver. Wang et al. [75] used the data obtained from a LiDAR laser scanner in order to correct the position localization faults generated by a GNSS localization system and a subsystem consisting of IMU sensors. Xiong et al. [76] proposed a modular localization system that uses image data, data collected from IMU sensors, information about the motion of the vehicle, and data from a GNSS receiver. Bresson et al. [77] followed a modular localization approach in which a decision layer fuses the information generated by a laser scanner-based SLAM algorithm, odometric lane tracking data extracted from color images in the RGB color space captured by a camera, and GPS position data. As a final example, Vishal et al. [78] made use of stored historical visual data that capture particular scenes in the environment and introduced a data fusion strategy that incurs a feedback loop. In particular, the authors proposed a method to harness GPS data by exploiting the image data. Moreover, they improved the localization-provided image data by extracting the information from GPS-based localization estimates. In terms of localization using only GPS data, the localization is improved by means of image retrieval over previously captured historical visual data.

The localization of FRs is much closer to the application of localizing pedestrians. This is due to the fact that the human walking motion is very likely to be close to the walking patterns observed in the FRs operating at a scene. It should be stressed, however, that the human motion data that an FR can generate may exhibit particular differences compared to the case of walking pedestrians. The irregularities in the observed motion data could be exploited by a localization method specifically targeted at FRs. Rantakokko and collaborators [79] discussed soldier and FR cooperative localization in indoor environments. Many studies target the localization of FRs (for instance, see [80,81,82,83]); however, very few of them propose modular localization architectures involving the fusion of outputs by at least two localization modules. At this point, we shall iterate the fact that FRs naturally operate in both indoor and outdoor environments, and they can commute from indoor to outdoor scenes and vice versa.

Now, we briefly review some modular localization systems that are targeted at pedestrians. In their study, Chdid et al. [84] combined an IMU approach that uses an accelerometer and a gyroscope sensor. The observations generated by these two sensors are fused by means of an extended Kalman filter, which outputs a position estimate. To complement the modularity of their system, the authors also integrated a vision-based component. This latter subsystem generates localization estimations via a pipeline that performs structure from the motion from the observed images. The output of both modules is averaged. This is a low-level fusion operation. Finally, the GPS and heart rate sensor data are combined in order to compute a more robust position estimate. Anacleto et al. [85] presented a specialized modular architecture for pedestrian localization. The system incorporates an inertial measurement system with sensors attached to both the foot and waist, including accelerometers, gyroscopes, and additional sensors. GPS and heart rate sensors augment the data to correct the inertial measurements, facilitating low-level fusion. Ultimately, GPS data, heart rate sensor data, and processed signals are integrated into a high-level fusion module to produce the final location estimation. Liu et al. [86] used an ultra-wide-band-enabled sensor and introduced a SLAM-based visual processing algorithm. They combined the data generated by both subsystems and then used an extended Kalman filter that estimates a pedestrian’s position. By means of this architecture, the authors tackled the scale ambiguity problem caused by monocular vision. Ali et al. [87] presented an indoor localization method that combines inertial measurement units (in particular, an accelerometer and a gyroscope) and an ultra-wide-band-enabled sensor. The authors integrated the information extracted from the raw positioning data from both modules by means of an extended Kalman filter. In their study, they also explored and analyzed the use of an adaptive Kalman filter for position prediction. According to the authors, their pipeline can fix position and orientation drift. In this case, the drift can be caused by the sensor interference and orientation dissensus originating from the inertial sensors. Chen and Hu [88] provided in their work a rigorous mathematical modeling analysis of how to combine inertial data and GPS location signals. The inertial measurements that they considered are generated by an accelerometer and a sensor that captures the angular rate of the foot. Their method is suited to outdoor localization. Huang et al. [89] introduced a unique methodology in which they employed inertial sensor measurements, together with Bluetooth and a light sensor, for human localization. The method proposed by the authors has four modules, namely a “data acquisition” module, a “motion model” module, a “light model” module, and a “decision making” module. The data acquisition component collects the motion data from inertial sensors and the light sensor, capturing positional human commute events. In the design of this last method, the decisionmaking module performs a higher-level analysis of the output of the modules by means of step-length modification, sub-edge heading reset, and pedestrian position revision.

In most of the solutions above, the different inputs for predicting the location are fused in one algorithm, such as a SLAM or another sensor fusion approach, which inhibits the benefits that some of the separate approaches might have under different conditions (e.g., GNSS solutions outdoors operate much better than indoors and, therefore, when outdoors, it makes no sense to fuse the sensor data with IMU data). Therefore, in this paper, we try to take advantage of separate technologies that perform differently in different environments and fuse them at a higher level, utilizing them in the best possible way in the environment they operate best in. This approach provides modularity and robustness to the system, allowing all the separate tools to operate independently and retrieve the estimated location that is more suitable for the environment in which an FR is operating in real time.

## 3. Methodology

In this section, we describe the overall methodology of the developed system. First, we provide an overview of the deployment platform in which the tools are integrated, including an overview of them, their main components, and how they interconnect. It is followed by the extended description of the three single tools: GNSS, visual, and inertial localization. Finally, in the last subsection, the details of the implementation of the Fusion tool that combines the three previous modalities are explained.

### 3.1. System Architecture

Here, we present the general architecture of our proposed multi-modal self-localization tools. In Figure 1, the architecture of the localization system and its integration with the deployment platform can be seen in detail. First, it can be seen that each individual tool has a dedicated hardware component, typically a sensor device or an integrated circuit board, apart from software that processes the input received by the external sensors. In the case of the Galileo-assisted self-localization, the main hardware components consist of a small protective case enclosing a processing unit, in this case a Raspberry Pi, accompanied by an external GNSS antenna. Accordingly, the hardware of the Visual self-localization tool consists of a helmet-mounted camera, which streams video in order to be used for the estimation of the movement. Finally, the hardware of the inertial self-localization tool consists of a smartphone integrating IMU sensors acting as the primary source of localization, as well as an auxiliary set of GNSS and BLE modules. On the contrary, the fusion self-localization tool does not have a dedicated hardware component, and it is a purely software tool utilizing the output of the other three self-localization tools.

In order for all our individual tools to communicate and for our proposed Fusion tool to receive the individual predicted locations from each tool, a powerful yet lightweight laptop was utilized that incorporates a suite of communication and orchestration between the individual tools. More specifically, it provides a wireless local area network between the tools and also has a Message Queuing Telemetry Transport (MQTT) broker [90] that allows each tool to share their predicted location with the other tools, including our proposed Fusion tool. Taking advantage of this implementation, the localization tools are able to retrieve information from one another, such as the visual-based and the inertial-based tools that obtain some calibration and correction parameters from the Galileo one. Likewise, the broker is key for the Fusion self-localization tool, which collects the localization information from the individual tools. The tools are set to generate location estimations roughly every 0.5 s, providing latitude, longitude, altitude, and heading of the FR. Some of the tools can provide extra information like the indoor/outdoor flag of the Galileo tool, estimation quality metrics, and sensor placement attributes. The use of the MQTT broker makes our solution modular, enabling any kind of visualization device to subscribe to the broker and consume the published location in order to be displayed to the FR or to the command center. Within this project, the FR visualization module relies on a helmet-mounted augmented reality (AR) display able to project useful operational information. However, this is not restricted to it, and, for instance, a mobile app or a command center terminal can also be used while it is connected to the local network.

### 3.2. Visual Self-Localization

In this section, the details of the implementation of Visual self-localization are presented. One important requirement of our solution was to be able to operate without, prior to operation, having any maps, floor plans, or models of the operational area. However, we would like our tool to be able to take advantage of such information, if it is available (i.e., a SLAM model of the operational area/building). In that manner, we opted for a solution based on monocular SLAM that can operate both with or without the pre-acquired model of the area.

SLAM is an ensemble of techniques, utilized mainly in robotics, that enables a device to map its surroundings and determine its own position simultaneously. Its aim is to help autonomous systems navigate unknown or changing environments. Although typically involving multiple sensors like cameras, LiDAR, and the GPS, our system focuses solely on visual SLAM due to hardware constraints and the need for the individual sub-components to operate independently. Monocular cameras inherently suffer from scale ambiguity, complicating accurate scale determination. This can lead to scale drift, where minor errors accumulate over time, affecting position estimations. However, advancements in monocular SLAM algorithms, such as loop closure, mitigate these issues. Loop closure recognizes previously visited locations, correcting errors and improving map consistency by merging similar features across different parts of the environment.

We chose to base our proposed solution on the ORB-SLAM2 [91] method. It utilizes FAST (Features from Accelerated Segment Test) [92] detector for the visual key point identification, which is then used by BRIEF (Binary Robust Independent Elementary Features) [93] descriptors for representing the key points detected efficiently. The matching of key points between consecutive frames is based on Hamming distance, enabling continuous tracking and map building.

From a software standpoint, the Visual self-localization tool is arranged into two modules: the outer shell and the localization core. In the flow diagram depicted in Figure 2, the localization core’s workings correspond to the “Estimate camera pose” stage, while all the rest correspond to the outer shell of the architecture. Internally, the localization core assumes a local coordinate system. Thus, a transformation from the local coordinate system to the global coordinate system must occur before formatting the output message to the agreed-upon format and sending it via the message broker to the rest of the system. The tool, given the appropriate configuration, can utilize either a perspective or a panoramic camera. The connection to the camera is handled by a UDP proxy server, which constantly reads the camera and exposes its stream to the core module via a network interface. In the localization core of the tool lies a modified version of the Stella-VSLAM C++ library [94], which consumes the exposed camera stream and communicates its output to the outer Python shell via a file-based UNIX Domain Socket (UDS). The outer shell constantly receives the camera pose expressed in local units and transforms it into a global GPS position utilizing calibration data in a way described in the following paragraph. Then, various derivative metrics like distance and heading are being calculated, and all the information produced is formatted into a JavaScript Object Notation (JSON) message before being sent to the message broker, closing the cycle.

The tool is able to perform localization without necessarily having traversed/scanned the operational area before. To achieve this, it needs to run upon instantiation, a calibration procedure to map the underlying raw SLAM model to reality. If we wish to use the tool instantly, without conducting calibration after instantiation, it needs to have the calibration data (the raw SLAM model and the local-to-global point correspondences) beforehand. In either case, the procedure in which the tool can calculate this local-to-global point transformation is hereby described:

We implemented a procedure that requires three points of known world coordinates to calculate the required transformation. To proceed, the user has to walk through all three points and notify the tool upon their arrival at each of them.

The world coordinate system represents latitude–longitude location points as 2D vectors on a curvilinear coordinate system that approximately fits the spherical shape of Earth. Let us define by $\bar{P}={\left(\begin{array}{cc} P_{x} & P_{y} \end{array}\right)}^{⊤}$ a point in the local 2D Cartesian coordinate system of the underlying SLAM model. Let us also denote by $P_{h}$ the height of point $\bar{P}$. To map point $\bar{P}$ to the corresponding point ${\bar{P}}^{\prime }$ in the world coordinate system, we trivially define a homogeneous coordinate transformation formula. First, define three calibration points $Q_{i}=\left(\begin{array}{cc} Q_{i}^{lat} & Q_{i}^{lon} \end{array}\right)$ for i∈{1,2,3} in the world coordinate system. Also, consider the respective Cartesian 2D calibration points in the local coordinate system of the underlying SLAM model, namely $Qc_{i}=\left(\begin{array}{cc} Qc_{i}^{x} & Qc_{i}^{y} \end{array}\right)$ for i∈{1,2,3}. For the pairs of points $Q_{i}$ and $Qc_{i}$, respectively, we also denote their associated altitude values by the scalars $Q_{i}^{alt}$. This pair of location point triplets models the association between points in the two location point vector spaces, and can help map points from the first space to the second space. To allow for this mapping, we define the homogeneous coordinate transformation matrices L and C; the former for the global world coordinate system and the latter for the Certesian system of the localization tool
(1)$$L=\left(\begin{array}{ccc} Q_{1}^{lat} & Q_{2}^{lat} & Q_{3}^{lat}\\ Q_{1}^{lon} & Q_{2}^{lon} & Q_{3}^{lon}\\ 1 & 1 & 1 \end{array}\right),\;C=\left(\begin{array}{ccc} Qc_{1}^{x} & Qc_{2}^{x} & Qc_{3}^{x}\\ Qc_{1}^{y} & Qc_{2}^{y} & Qc_{3}^{y}\\ 1 & 1 & 1 \end{array}\right)$$

To map the points from the Cartesian space to the world coordinate system defined by the three points $Q_{i}$ in this latter space, we define the linear operator T as $T=LC^{-1}$. Then, the projected point ${\bar{P}}^{\prime }={\left(\begin{array}{cc} P_{lat}^{\prime } & P_{lon}^{\prime } \end{array}\right)}^{⊤}$ in the world coordinate system, given the associated point $\bar{P}$ in the local coordinate system, is provided by the matrix–vector product
(2)$${\left(\begin{array}{ccc} P_{lat}^{\prime } & P_{lon}^{\prime } & 1 \end{array}\right)}^{⊤}=T\cdot {\left(\begin{array}{ccc} P_{x} & P_{y} & 1 \end{array}\right)}^{⊤}$$

We are also in need of approximating the altitude value α of point ${\bar{P}}^{\prime }$ (α should not be confused with the symbol $P_{h}$ representing the height of point $\bar{P}$), or in equivalence of point $\bar{P}$. We approximate α by the formula
(3)$$\alpha =\frac{1}{3}\times \left(Q_{1}^{alt}+Q_{2}^{alt}+Q_{3}^{alt}\right)-P_{h}\times s$$
where *s* is a scaling factor computed by the formula
(4)$$s=D\left(Q_{1},Q_{2}\right)/\sqrt{{\left(Qc_{1}^{x}-Qc_{2}^{x}\right)}^{2}+{\left(Qc_{1}^{y}-Qc_{2}^{y}\right)}^{2}}$$
and D(·,·) calculates the geodesic distance (computed in m) of the 2D points, $Q_{1}$ and $Q_{2}$, in the 2D latitude–longitude space. The denominator in the fractional representation of the scale factor *s* in Equation (Equation 4) is the Euclidean distance of the calibration points $Qc_{1}$ and $Qc_{2}$ in the local coordinate system.

The aforementioned calibration points can be either defined prior to the execution via command-line arguments or be chosen on the fly, during the execution as the calibration is being conducted, querying their world coordinates in real time by the Fusion localization tool (functioning during that time utilizing the rest of the self-localization tools). As we mentioned earlier, the tool has to be notified upon the arrival of the user at the chosen calibration points. This can be accomplished either by presenting a special QR code to the camera to be recognized or by sending the tool a configuration message from an auxiliary Android application we developed.

### 3.3. Galileo Self-Localization

Our proposed Galileo-assisted Localization Tool (GLT) is a multi-band and multi-frequency GNSS-based approach that utilizes a high-precision receiver with a dedicated active antenna. All data processing is performed on a small single-board computer (SBC), making the entire system self-reliant and capable of being powered by a small battery for several hours. One of the primary focuses during the development of our GLT is tailoring it to the needs of FRs in terms of size, weight, and robustness. To prototype the system, we opted for one of the smallest breakout boards available, the SparkFun GNSS receiver board, which features a U-Blox ZED-F9P receiver chip [95]. For the SBC, we selected a Raspberry Pi Zero 2W, chosen for its compact size and capable processor unit, allowing it to receive and process GNSS data at 2 Hz and wirelessly transmit the data via WiFi to the orchestrator. The GNSS receiver board is connected to the SBC via UART (Universal Asynchronous Receiver/Transmitter). We employ a small GNSS active patch antenna with a low-noise amplifier to receive signals from common GNSS systems such as BeiDou, GLONASS, Galileo, and the GPS at two frequencies.

Figure 3 provides a high-level view of the GLT design architecture. A dual-frequency GNSS refers to tracking multiple radio signals from each satellite on different frequencies. For the GPS, this involves L1 and L5, and, for Galileo, E1 and E5a. A significant advantage of dual-frequency transmission from a single satellite is the ability to directly measure and eliminate ionospheric delay errors for that satellite. Additionally, receivers with dual-frequency support can better distinguish between direct and reflected signals compared to single-frequency receivers. In practical terms, dual-frequency capability increases the accuracy to approximately 1 m, compared to several meters for single-band receivers. Dedicated tests in previous research (citation removed for review) have compared the accuracy of current dual-frequency smartphones with our proposed GLT. While the best performance with smartphones is worse than 3 m, our GLT solution delivers on average 1.4 m or better. While such high accuracies may not be essential for most rescuer needs, they are often crucial in challenging environments such as operations near collapsed or intact buildings or during assessment walks in wooded areas.

For practicality, we have developed two physical versions of the GLT. One can be carried inside a pocket in the personal protective equipment (PPE), while the other is integrated into an FR’s helmet. All components, including a 2000 mAh power bank, fit into a small box measuring 75×50×33 mm, or alternatively on the backplate of the F2XR Gallet helmet. The active antenna should be mounted as high as possible and face the sky, ideally positioned in the breast or arm pocket, or under the front plate of the helmet. On the software side, the GLT integrates custom adaptations to enhance performance and streamline data transmission. At the receiver level, modifications have been made to achieve a higher update rate of 2 Hz, doubling the default frequency. Additionally, enhancements have been made to improve the precision of the coordinates, resulting in an order of magnitude increase compared to standard configurations. This adjustment enables more frequent and precise location updates. Furthermore, the Raspberry Pi Zero 2W serves as the computational hub, running custom Python scripts tailored to handle incoming NMEA (National Marine Electronics Association) and U-Blox messages from the GNSS receiver. These scripts decode and parse the raw data, extracting essential information such as coordinates, accuracy metrics, and heading. Leveraging the processing power of the Pi Zero 2W, real-time calculations are performed to ensure accurate positioning data are continuously available. The GLT starts to find a position fix after it is turned on. It sends position data immediately after there is a fix. It should be noted that it only sends data if the accuracy of the tool is below 10 m. We introduced this restriction because otherwise it would send positions that are erroneous for this use case. To obtain a position fix, it can take up to a minute. The tool should be started outdoors; otherwise, no data will be sent because in most cases indoors it cannot obtain a position fix. If the tool is once started outdoors, achieving a position fix again after being indoors is faster. Once processed, the extracted information is transmitted wirelessly to the data-sharing orchestrator using established communication protocols. This seamless integration between hardware and software components enables efficient data flow and facilitates rapid dissemination of critical location data to relevant intra-team members and command and control forces.

### 3.4. Inertial Self-Localization

Our proposed inertial-based self-localization (INERTIO) tool is a sensor fusion-based approach that uses sensors onboard smartphones and applies a collaborative and opportunistic method to reduce drift due to the inherent noise of IMU-based sensors. The main component of INERTIO is providing a location prediction and orientation of the FR using the onboard IMU sensors in combination with using other modalities, such as collaborative information from other mobile phones and opportunistic information from known locations (e.g., landmarks, beacons, etc.).

Figure 4 provides a high-level view of INERTIO design architecture. More specifically, it details the various sensor modules to be utilized and fused of an Android-based smartphone to obtain good-quality self-positioning estimates. In line with current state-of-the-art approaches, INERTIO will use the accelerometer, gyroscope, and magnetometer of the smartphone device’s onboard IMU sensors as the primary basis for obtaining the initial position and orientation of the FR. Furthermore, satellite signals of the GNSS from a GLT or the GPS from a smartphone are collected to obtain a GPS location to be used to enhance and correct the predicted position from the IMU sensors. Also, any other additional external data sources available (e.g., landmarks) are to be incorporated into the INERTIO positioning algorithm to further enhance and correct the predicted position of the FR. Lastly, the location and trajectory information derived is outputted via WiFi to be distributed to other mobile phones.

Figure 5 showcases how the onboard IMU sensors (on the left-hand side) and other data modalities (on the right-hand side) available on the smartphone device are utilized by INERTIO positioning model. The accelerometer is used to detect when the FR has moved (i.e., taken a step), and then with these acceleration measurements to derive the stride length of that step to provide a new initial IMU-based position. However, before accelerometer readings are used, an extended Kalman filter (EKF) from the FSensor library is utilized to denoise the raw acceleration measurements and thus grant enhanced stability. Regarding orientation estimation for the initial IMU position of an FR, this is now conducted by a Madgwick filter, which primarily uses gyroscope readings in tandem with accelerometer and magnetometer readings to derive an orientation. Then, as before in the initial model, this orientation information is used in conjunction with the distance traversed by the FR from the stride estimation to advance the FR to the next estimation IMU-based position. This IMU position can then be corrected opportunistically from additional data sources, such as other mobile phones or landmarks with known locations. More specifically, in INERTIO, we utilize the onboard BLE module of an FR smartphone device to provide additional information to help correct location drift being experienced. This is achieved by first exchanging the current positioning information (latitude, longitude, heading, drift, etc.) with other smartphones running INERTIO connected to the same network using the existing MQTT service. More specifically, this ensures that every smartphone device running INERTIO knows the current position of every other smartphone. Secondly, each smartphone device is continuously broadcasting (at a rate of approximately one packet per second) BLE advertising data packets containing a unique set of identifiers while simultaneously listening for such packets broadcasted via BLE. Therefore, when two or more FRs move into proximity of each other, these advertising packets are received in which the BLE RSS is measured [96] and from which the approximate distance between them can be calculated. Then, this measured distance can be compared against the predicted distance derived using the exchanged positioning information of that FR and their own predicted position. Lastly, the difference between the measured distance (from BLE) and the predicted distance (from current positioning information) denotes the amount of location correction of that FR that may need to take place. The amount of location correction to be taken is determined by comparing their own current potential drift to the potential drift of the device from which the BLE advertising packet originated, with a larger or smaller correction taking place if the drift of the other FR is smaller or larger than their own, respectively. Furthermore, in some cases, it is expected that a building is equipped with devices that have known fixed locations, thus providing a landmark/anchor for indoor positioning purposes [97]. In such cases, these devices that include a BLE module broadcast BLE advertising packets similar to those used in collaborative correction in which each packet contains a set of unique identifiers to distinguish one device from another. Again, INERTIO is continuously listening for these BLE advertising packets, and, when received, the distance between the smartphone and device is calculated using the RSS of the received packet. However, as the location of these landmarks is known and does not change, it is possible to then set the predicted location of the FR to match that of the landmark displaced by the measured distance to help compensate for any drift that may have occurred.

### 3.5. Fusion Self-Localization

The primary aim of this work is to combine the output of all three individual tools, providing a robust and fail-safe option for self-localization based on the sensor fusion principle. After experimenting with Kalman-type filters [98], we soon realized that such an approach would not be appropriate for our use case. By design, Kalman filtering processes require a priori knowledge of the expected noise statistics. This would entail conducting generic experiments with ground truth positions to try to estimate their generated noise distribution and obtain constant values to be used as noise. However, this approach is not suitable for the dynamic environments where our tools operate and the nature of our tools.

More specifically, the noise characteristics of our three tools vary significantly, not only across individual runs but also as they evolve in time through a single run. For example, the Visual localization tool is known to produce models with sometimes varying scale across its different sections. After calibration, this leads to a skewed model, whose inducted noise is relative to the distortion of the current region of the model where the FR is located. As a consequence, having reliable and generalized noise statistics is challenging. Another behavior hindering the extraction of reliable noise statistics is the fact that INERTIO can incorporate position corrections from different available sources in an opportunistic way, if and when available. This means that its inaccuracy/noise is not predicable as it depends on different modalities that might be available in different situations (e.g., GNSS or landmark availability) where a sudden correction of its position could be performed, altering/resetting the noise characteristics of the tool. Lastly, even the noise of the GLT can vary between urban, suburban, and rural areas (or even between different regions of the globe), raising the need for at least three separate noise models to correct its behavior.

Due to these reasons, a Kalman filter-based solution was considered impractical and we turned towards a heuristically driven decision tree approach. Based on strong evidence gained throughout experiments conducted during pilots involving real FRs, regarding the behavior of our tools, we were able to draft a tool selection algorithm that is reasonably expected to choose the best out of our three localization tools for any given scenario. Our strategy is roughly to collect and retain the messages from all the available tools published to the broker recently (being no older than two times the agreed update interval of the tools—in our case, messages no older than one second) and then re-transmit the best of them adhering to the following algorithm.

First, if there is a recent GLT message available, with a reliable accuracy and with the indoor/outdoor flag suggesting that we are outdoors, the Fusion tool selects GLT since it is the most reliable and practical in this setup. We base the detection on whether we are indoors or outdoors on information provided by the GLT tool since the GNSS signals will only partly or not at all be received indoors. For indoor/outdoor detection, we investigated different attributes of the GNSS receiver: the horizontal accuracy provided by the U-Blox device itself, the number of visible satellites, number of satellite vehicles used for GNSS calculation, and the signal-to-noise ratio of the respective satellites. We concluded that the most reliable attribute to tell whether the person is indoors or outdoors is the horizontal accuracy. It clearly shows higher values for indoor locations and low values when we are outdoors. As a consequence, if the horizontal accuracy is below a certain threshold (for this series of experiments, we used a threshold of 2 m), we assume that we are outdoors and indicate this in the indoor/outdoor flag of the GLT message sent to the broker.

Second, if the GLT indicates that we are indoors and there is a recent visual localization message available, we utilize that. Favorable behavior of the Visual self-localization tool is that, when it loses tracking, for example, due to the bad lighting conditions or a series of very sharp motions, and thus cannot infer position, it does not emit any messages. Therefore, its last message will become obsolete soon enough and will play no role in the fusion selection procedure. Finally, if there is no preferable solution available, the algorithm resorts to using INERTIO as it always produces an output. When the Fusion tool receives a valid message again from a modality with higher priority, it switches back to that, continuing the operation in this iterative manner. The aforementioned strategy is better explained visually in Figure 6.

## 4. Results and Analysis

In the following paragraphs, we will present the results of an on-the-field experiment where a search and rescue operation was simulated involving one FR searching for victims in a mixed environment, including outdoor, sufficiently lit, and completely dark indoor spaces that were traversed in a predefined route (see Figure 7 right). Five runs of the designated route were conducted, comprising 37 waypoints (11 outdoor, 15 indoor in a well-lit area, and 11 indoor in a dark area) on which we measured the error of the location estimations. The waypoints are on average 2.73 m apart, giving the full route a total length of 98.37 m. The tests were carried out at the premises of the Navacerrada Fire Station in the administrative region of Madrid, Spain.

The tracing path begins outdoors, in the area surrounding the main facilities. In this phase of the experiment, the Fusion tool is expected to utilize the GLT modality as its primary source of localization. Moving to the next phase, the FR enters a small bunker consisting of three rooms and a connecting hallway. As we move indoors, the GLT begins experiencing degraded signal reception and raises the “indoor” flag to the system, just before completely halting emissions as it loses satellite fix. Being in that state, the hallway and the far room (the first one to visit) are sufficiently lit, and, as a consequence, the visual modality is expected to take over. As the scenario progresses further and the remaining two rooms lack any window or artificial source of lighting, the Visual tool is expected to lose tracking and stop emitting as well. When this happens, the Fusion algorithm resorts to INERTIO as its only source of positioning under these harsh conditions. Gradually, the FR returns again in the sufficiently lit hallway before coming out of the bunker, concluding the run outdoors. The above-described routine takes on average 262.2 s to complete and triggers six source modality transitions for the Fusion self-localization algorithm.

### 4.1. Ground Control Points for Evaluation

To validate our proposed fusion approach, we conducted surveys to establish ground truth through two distinct methods. Outdoor ground control points (GCPs) were surveyed using a real-time kinematic GNSS, achieving a mean horizontal accuracy of 1 cm. The designated outdoor GCPs are numbered 1 to 7 and 35 to 37 and distributed near the building used for the experiments. Indoor reference points were established based on an IPS (refer to Section 2.4) trajectory obtained during an inspection run. In the confined bunker space, GCPs 8 to 34 were selected to provide comprehensive indoor coverage. During the IPS survey, each indoor GCP was carefully marked, and the IPS device was precisely positioned at each point. The IPS utilizes inertial measurements, visual odometry techniques, and a stereo camera system for 3D orientation and position estimation, generating a real-time geo-referenced trajectory calibrated over outdoor GNSS-surveyed GCPs. However, it is important to note that the accuracy of indoor reference points, estimated at approximately 30 cm per point, is lower compared to outdoor GNSS points due to the IPS’s decreasing accuracy over time. This value is attributed to the closed-loop error of the IPS system, which we have determined to be around 30 cm. We acknowledge that achieving higher accuracy indoors would have been possible using tachymeter or total station survey equipment; however, for our specific case, this level of precision was not necessary and would have required significantly more effort. It is worth mentioning that, during the IPS survey, we also obtained a detailed 3D model of the operational area as a beneficial byproduct (see Figure 7 left).

To capture and record the time at which user visits each GCP during each experimental run performed, a stand-alone mobile application was developed that was used by a nearby operator. More specifically, this application allows the operator to input the current GCP number occupied by the user, which is then sent via the MQTT broker to the laptop carried by the user. These messages along with the outputted data of each localization tool are then logged on the laptop, ensuring synchronization of timestamps and enabling data analysis to be performed after the completion of the experiments. To ensure that each point was reliably captured and logged, when the user visited each GCP, they would pause (i.e., stand still) for 1 to 2 s.

### 4.2. Experimental Setup

An actual view of the experimental setup is provided in Figure 8. The camera of the Visual self-localization tool, as well as the GLT antenna, are mounted on the helmet. The smartphone for INERTIO is strapped at the leg of the FR, and it also carries the GLT device and the laptop hosting the message broker, the Visual, and Fusion localization tools.

As regards the setup of the Visual self-localization tool, in order to save valuable time and speed up the processes, we chose to run the tool in the pre-calibrated mode. To facilitate this, we visited and traversed the indoor spaces of the operational area one hour prior to the tests, storing the derived raw SLAM model for later use. Immediately afterwards, we calibrated/mapped the raw SLAM model to reality by conducting the 3-point calibration procedure described earlier. As calibration points, we utilized the GCPs numbered 8, 14, and 17 along the tracing path.

The GLT was turned on and off before and after each of the 5 experimental runs. Upon activation, the tool quickly achieved the accuracy necessary to start broadcasting. This is facilitated by its initial exposure at a vantage point almost on top of the bunker with clear-sky visibility.

Regarding INERTIO, it was deployed on a Google Pixel 6 Pro smartphone device that was running Android version 13, which was securely fastened on the outside lower right leg of the user with the use of a phone-exercise strap. The tool was calibrated with the initial heading of the user, that is, in the direction of GCP 2 from GCP 1, and was also provided network configurations to allow communication with MQTT broker hosted on the laptop. Furthermore, inside each dark room, we made the assumption that there were two known positions that could be utilized as landmarks to allow INERTIO to correct its position when coming to a very close proximity with them (less than 0.5 m). For this, two Raspberry Pi 3B+ devices were deployed at GCPs 23 and 30 to act as BLE-based landmarks, which broadcast BLE advertising packets at a rate of 5 Hz. The coordinates (i.e., latitude and longitude) along with unique identifiers of each of these landmarks were passed to INERTIO in order for position corrections to be performed as the user comes into near proximity.

After the tools’ initialization was completed, we conducted five consecutive test runs with small breaks in between inside a time window of about one hour.

### 4.3. Results

The location error at the GCPs was assessed over five test runs. Figure 9 illustrates the distribution of the mean location error, with the error bars indicating the measurement variance. In the upper section of the chart, insight into the source modality transitions triggered during the traversal of the full route is provided. Table 1 provides a summary of the location error for all the self-localization tools, taking into account only the location estimations utilized by the Fusion tool in each run. On the other hand, Table 2 presents the location error for the individual localization modalities, considering the partial availability of the tools at the various GCPs. Specifically, the INERTIO tool demonstrates full availability at all 37 GCPs, indicating consistent data availability. The visual data are predominantly available in lit indoor areas where the SLAM model acquisition was carried out. Note that we could set the Visual self-localization tool to also provide estimations outside of the acquired SLAM model but deemed it unnecessary due to the given availability of the other two localization modalities. The GLT data, on the other hand, are primarily available outdoors. Lastly, Figure 10 showcases the individual tools’ location error values at all the GCPs, plotted together with the Fusion tool’s actual error and the theoretically optimal value had it chosen the correct source modality at all times. Indeed, in the frame of our study, the Fusion self-localization tool seems to make the optimal modality choice at 35 out of the 37 GCPs, where at all times there are at least two modalities available for use. The two GCPs where the Fusion tool fails to pick the most accurate modality are the transitional ones (7 and 8), right upon entering the concrete building. This indicates a reactance on behalf of the selection mechanism to release the GLT and turn sooner to the Visual localization tool, behavior related to the indoor/outdoor detection mechanism. Before we delve into the analysis of the results, we should underline the fact that all the measurements were taken at the position of an FR upon arrival at the designated ground control points, and, as a consequence, there is some tolerance to their precise position due to their perception and body pose, unavoidably adding some noise to the results.

Starting with the contribution of the GLT to the fused setup, the results of our test runs reveal an average GNSS localization accuracy of 1.73 m. This slightly deviates from our previously reported average accuracy of 1.4 m, which was derived from long outdoor runs (citation removed for review). In the current case, the test scenario involved a shorter run with close proximity to a building, significantly affecting the GNSS signal reception. Particularly noteworthy is the sharp transition in accuracy upon entering and leaving the building, suggesting strong interference from the nearby structure. Furthermore, the limited time available for satellite signal acquisition upon exiting the building likely contributed to the incomplete or partially ambiguous signal reception. However, as the test progressed beyond the immediate vicinity of the building, we observed a gradual improvement in accuracy, with the subsequent reference points yielding accuracies of 1.9 m and 1.8 m, respectively. These findings underscore the impact of environmental factors on the performance of the GNSS.

Continuing with the indoor/outdoor detection provided by the GLT that lies at the heart of our Fusion algorithm, we observed that the tool sometimes provides good horizontal accuracy values even if the actual positioning is bad. The reason behind this behavior seems to be that the NLOS signals are interpreted as LOS signals, indicating that the tool suffers from multi-path effects causing some inconsistency on the indoor/outdoor transition detection mechanism. In addition, and as mentioned above, the tool needs some time to obtain a position fix and high accuracy after an indoor–outdoor transition. This results in late outdoor detection and INERTIO output being used instead when coming out of the building. Keeping in mind that the accuracy of the GLT is degraded a while after being indoors (see also Figure 11); this “takeover” from INERTIO is reasonable and enhances the overall fusion accuracy.

When the FR approaches the entrance of the building, the Visual self-localization tool starts picking the features of the pre-acquired SLAM model and begins emitting location estimations. As long as the GLT emits estimations tagged with the “outdoor” flag, it is preferred by the Fusion tool, so we need to advance further inside the building for the Visual tool to take over, as can be seen in Figure 9. Specifically, GCP number 8 in four of the five runs was localized with the Visual self-localization tool and in one run with the GLT. When well inside the building, in the lit rooms, the Visual tool appears to function at its best, showcasing a mean error of 37 cm, notably low considering the tolerances of the indoor GCPs’ ground truth positions and the added noise by the imprecise placement of the user on top of them. Note that the Visual self-localization tool successfully recovered the tracking at all times when the FR moved outside the dark rooms, enabling the correct and swift transition of the Fusion tool to it.

The results for INERTIO showcased in Figure 9, highlighted in green, are typical of a mobile IMU-based positioning approach, in which higher amounts of inaccuracy are being experienced due to pedestrian dead reckoning drift taking effect. A notable amount of IMU drift affecting INERTIO when it is first used by the Fusion tool (at the start of entering the first dark room in GCP 20) was expected due to approximately 54.6 m of movement being performed beforehand. At GCPs 23 and 30, we can clearly observe the user moving in close proximity to their respective BLE landmarks and a positioning correction being performed, with the location error in both instances dropping below 1 m. However, the location error rapidly increases in the subsequent GCPs, further highlighting the existence of IMU drift, more specifically, gyroscopic drift of the heading estimation, causing the user to be advanced in a non-optimal direction. This is further evident in Figure 11, in which we can observe the INERTIO output (green line) not moving around the two dark indoors rooms in a smooth elliptical fashion, in line with the sequence of the GCPs. Moreover, Figure 11 grants insights regarding how INERTIO is able to perform corrections based upon the outputs of the other localization tools. For example, we can observe that INERTIO is able to effectively fuse the location information outputted by the GLT for the first seven GCPs as the locations of each tool closely follow one another, as shown by the yellow and blue lines, respectively. This is also evident from the fact that the total mean error of INERTIO in all the location estimations (Table 2) is smaller than the mean error when we only consider the location estimations where INERTIO was used by the Fusion tool (Table 1).

## 5. Discussion

Our study investigated the efficacy of a novel approach to localization, which leverages complementary fusion techniques to improve accuracy and reliability. Building upon the existing literature, which predominantly focuses on sensor fusion at a low level, our approach operates at a higher level, integrating multiple localization modalities to produce more robust results. One of the key findings of our approach is the successful implementation of complementary fusion, where multiple sources of spatial information are utilized to improve the localization accuracy. By selecting the best option available from the fused sources, we were able to mitigate the errors and maintain accurate localization even under challenging conditions. Different fusion techniques that could handle a weighted combination of the tools when more than one is available, such as Particle Filtering (PF), may be explored in a future revision of the method. In addition to the technical aspects of our approach, the environmental conditions emulating real-world emergency scenarios added much to its value. Despite these adverse conditions, our localization system demonstrated resilience, effectively mitigated errors, and maintained sufficiently accurate position estimates, highlighting its potential utility in emergency response situations. Overall, our study contributes to the body of knowledge on sensor fusion and localization systems by presenting a novel approach that addresses the limitations of the existing methods and demonstrates practical applicability in real-world scenarios.

While our study has showcased the effectiveness of our novel approach in challenging real-world scenarios, there remain opportunities for further enhancement of the specific components. In our experiment, we did not exploit any correction data with the GNSS tool. Since we are using the same U-Blox device as for the outdoor GCP measurements, we are able to achieve similar accuracy to the GCPs if non-in-orbit correction data and a large antenna are used. However, we cannot assume receiving continuous correction data in emergency cases, such as via LTE. The integration of in-orbit and non-in-orbit correction data in the GLT is foreseen for future research. Especially, the use of in-orbit correction data is very promising but depends on the availability of the respective receiver hardware. The future work will also include a refinement of the indoor/outdoor indicator. The threshold, for instance, can be automatically adjusted, particularly for specific transitions and different antennas with varying mounting positions. Additionally, we will explore how different types of buildings affect the indicator. These adjustments will not only enhance the performance of the GLT but also directly impact the performance of the Fusion tool.

As regards the potential improvement of the Visual self-localization method, we believe that it should be addressed in two areas. The first would be to improve the robustness of the tracking by utilizing some newer feature extractor. This would enable the tool to utilize SLAM models that were acquired much earlier in time and with a much different lighting setup than the one present at the operation time. The second would be to minimize the scale drift that is exhibited in our monocular camera setup. To alleviate this, our proposal is to either invest in a scale correction mechanism based on common object detection and the prior knowledge of their true dimensions or migrate to a stereo dual-camera setup. Lastly, we plan to conduct a new series of experiments with the non-pre-calibrated configuration of the tool, creating and calibrating the underlying SLAM model on the fly utilizing the Fusion tool, emulating the operational conditions where no previous information of the area is available.

Finally, although INERTIO granted a reasonable level of performance, the observed gyroscopic drift can impact the heading estimation, especially after the user has moved longer distances, which is a long-standing problem in the scope of IMU-based positioning systems. This is particularly highlighted in the results (Section 4) when traversing the dark room sections of the performed experiments, in which the localization estimates are solely from INERTIO and the highest amount of location error was experienced. A future direction to help alleviate this is to consider the heading estimation outputted by the other localization tools as a means of correction, similar to the corrections performed by INERTIO for its position (i.e., latitude and longitude). Another enhancement towards improving the heading drift could be by deploying and utilizing additional heading estimation models, such as extended Kalman filter (EKF), and then performing either model selection or data fusion to further prolong and mitigate the effect of gyroscopic drift. Finally, in cases where multiple FRs are moving together in a search and rescue operation, in the future, we could explore how collaborative positioning can be utilized, where multiple devices in proximity performing IMU-based localization exchange positioning and heading information between one another in order to correct and better compensate for their own errors.

## Figures and Tables

**Figure 1 sensors-24-02864-f001:**
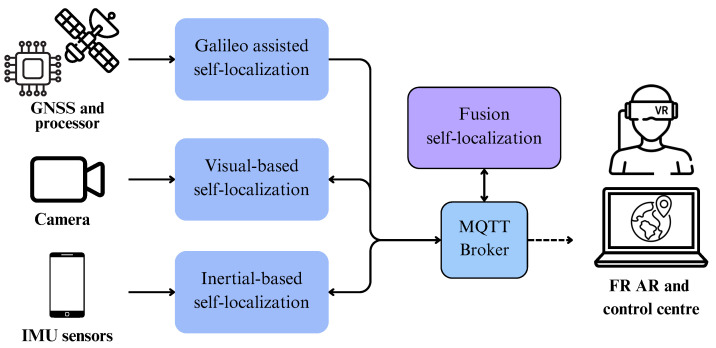
System architecture. On the left, the figure shows the dedicated hardware of each localization tool. The central part describes the intercommunication between modules through an MQTT broker. On the right side, the endpoints of the platform consuming the data are presented.

**Figure 2 sensors-24-02864-f002:**
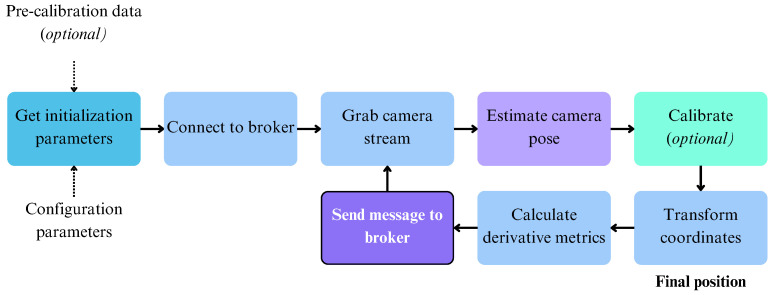
Visual self-localization tool flow diagram.

**Figure 3 sensors-24-02864-f003:**
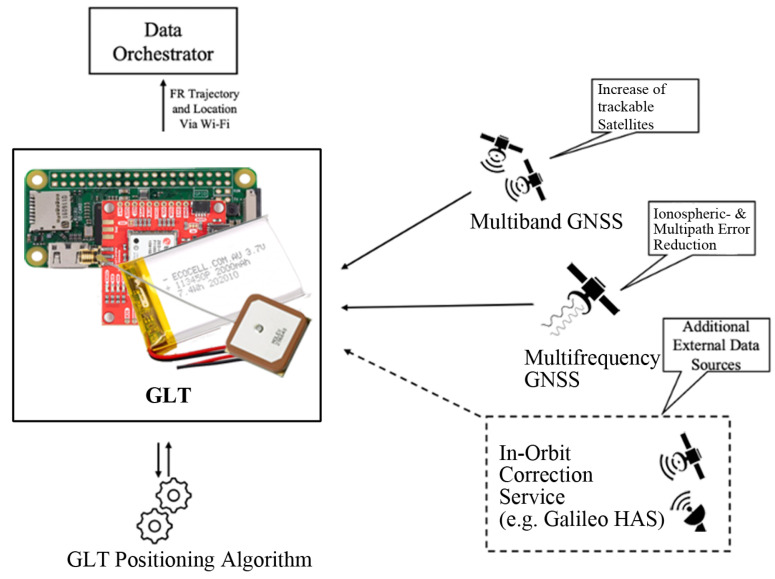
Galileo-assisted Localization Tool High-Level Architecture.

**Figure 4 sensors-24-02864-f004:**
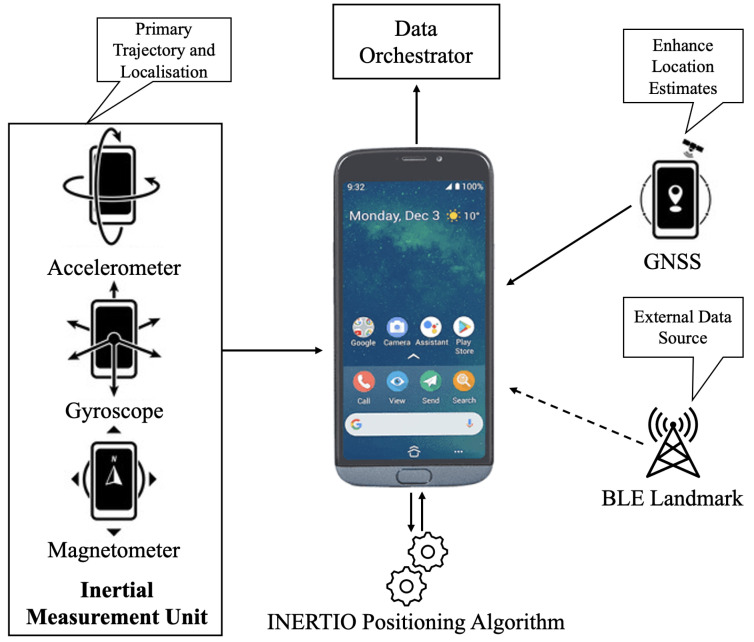
INERTIO High-Level Architecture.

**Figure 5 sensors-24-02864-f005:**
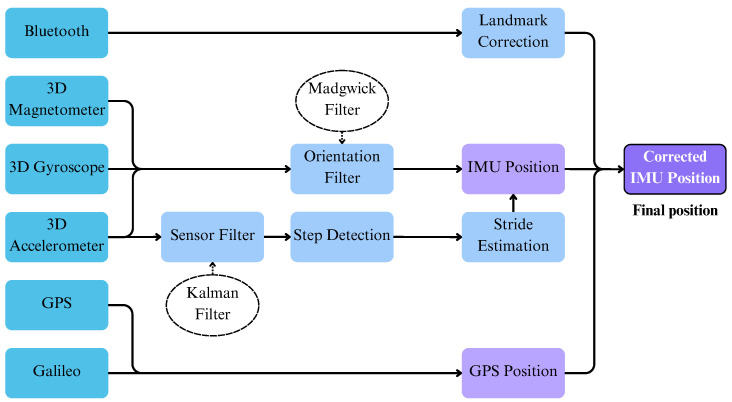
INERTIO Architectural Model.

**Figure 6 sensors-24-02864-f006:**
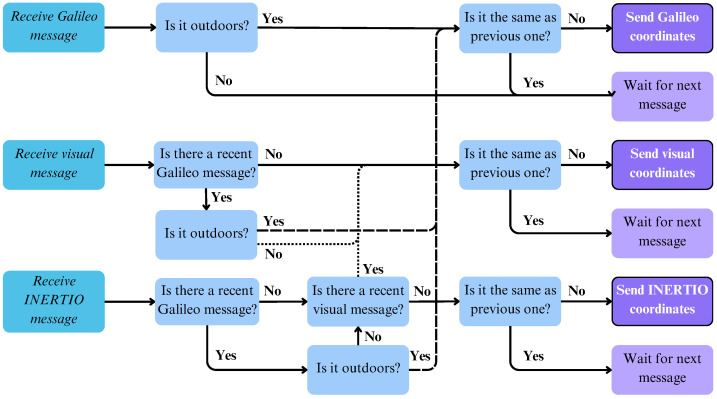
Flow chart of our fusion localization algorithm. Note that, by “recent”, we refer to a message no older than 1 s (twice the individual localization tools’ agreed update interval).

**Figure 7 sensors-24-02864-f007:**
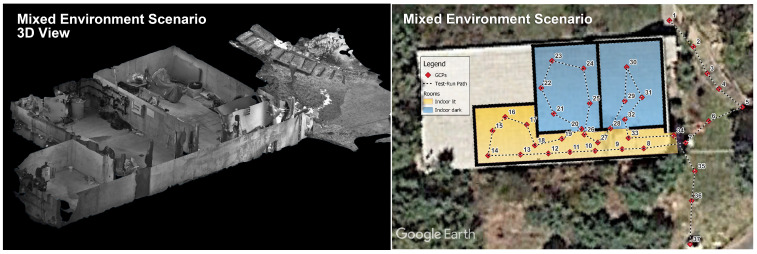
Setup of the scenario: indoor in 3D (**left**) and a 2D overview (**right**) with the ground truth path overlaid in dotted lines.

**Figure 8 sensors-24-02864-f008:**
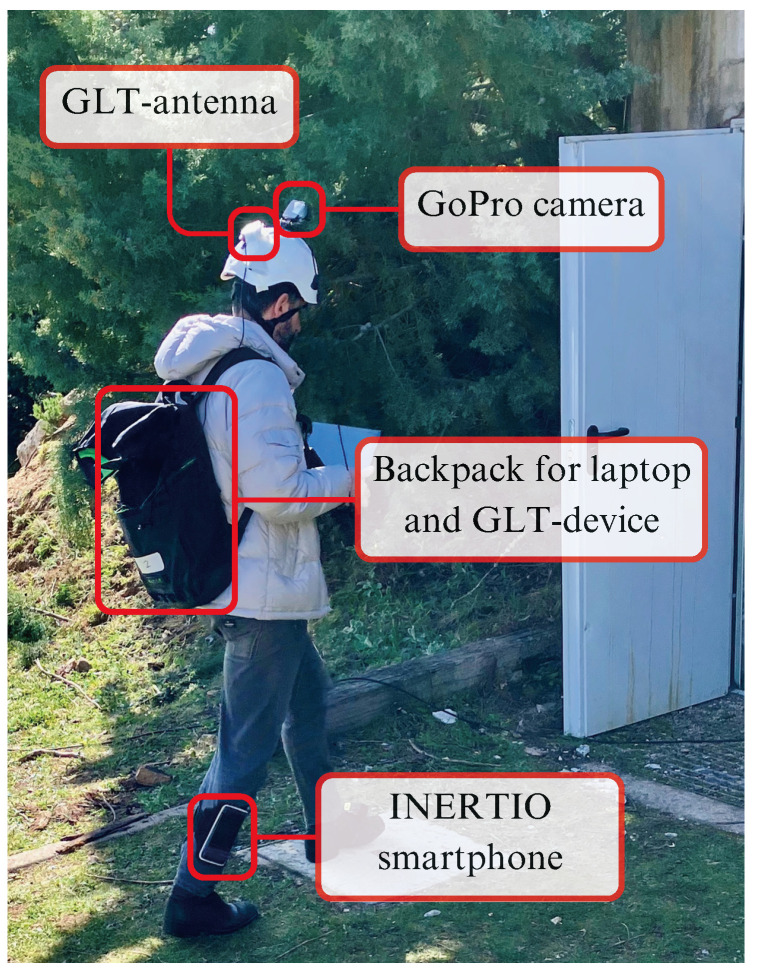
Hardware setup mounted on the user combining every self-localization module for the experiment.

**Figure 9 sensors-24-02864-f009:**
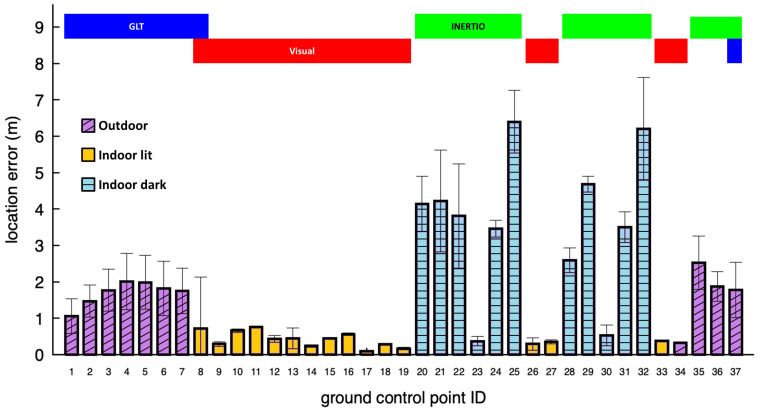
Location error at ground control points (GCPs). Five test runs were conducted. The overlaid error bars indicate one standard deviation around the mean values. The 37 GCPs are divided into 3 distinct sectors: 11 of them are situated outdoors, 15 indoors at a well-lit area, and 11 indoors at a dark area. Traversing the full route should trigger six source modality transitions. On the upper section of the chart, we can see the actual source modality used by the Fusion self-localization tool. Note that GCPs 8 and 37 were localized by more than one modality across the runs.

**Figure 10 sensors-24-02864-f010:**
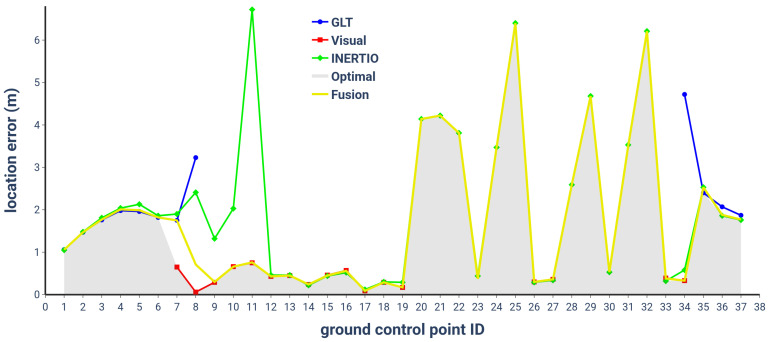
The average location error of all tools at GCPs. Not all tools are available at every GCP. The Fusion self-localization tool manages to make the optimal modality choice at 35 out of the 37 GCPs. Note that at all times there are at least two modalities available for use.

**Figure 11 sensors-24-02864-f011:**
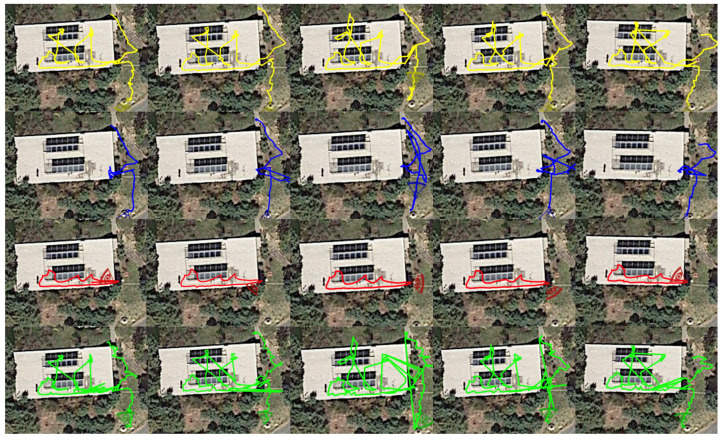
All self-localization tools’ outputs as visualized in the command center interface. From top to bottom, we can see the traces produced by Fusion (yellow), GLT (blue), Visual (red), and INERTIO (green) tools. The columns indicate the ID of the test run, from run 1 (left) to run 5 (right). As observed, the Fusion tool trace is a composition of selected pieces from all the other tools’ traces. Note that, in the 5th run, due to a temporary network malfunction, we lack the visualization of its beginning, as can be noticed by the late start of the Fusion, GLT, and INERTIO traces.

**Table 1 sensors-24-02864-t001:** Location error for all self-localization tools. For the GLT, Visual, and INERTIO measurements, we take into account only the location estimates that were actually used by the Fusion tool in the respective run.

Tool	Mean Error (m)	Std (m)	Min (m)	Max (m)
GLT	1.73	0.69	0.24	3.33
Visual	0.37	0.20	0.04	0.80
INERTIO	3.37	1.92	0.18	8.73
Fusion	1.74	1.79	0.04	8.73

**Table 2 sensors-24-02864-t002:** Location error for the three individual self-localization modalities. For the measurements, we take into account all the provided location estimates. Note that the INERTIO tool provided location estimates at all the GCPs. On the other hand, the GLT provided estimates only outdoors or slightly inside the building, while the Visual tool only indoors and right in front of the building where the SLAM model extended to.

Tool	Mean Error (m)	Std (m)	Min (m)	Max (m)	Availability
GLT	2.17	1.32	0.24	8.47	12/37
Visual	0.38	0.21	0.04	0.78	17/37
INERTIO	2.03	2.32	0.06	15.83	37/37

## Data Availability

All data generated in the experiments conducted and processed in the frame of this work is publicly available in the following repository: https://drive.google.com/drive/folders/1by9qgiUUt_KrbT5utMhVZtL-mjKiwQXL (accessed on 30 January 2024).

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
