# Peer review of "Seamless Fusion: Multi-Modal Localization for First Responders in Challenging Environments"

_sensors, 2024, doi:10.3390/s24092864_

Round 1

Reviewer 1 Report

Comments and Suggestions for Authors

General comment on the topic: The fusion of different position estimates / different methods is not new. Many applications have been investigated using KF and PF as base fusion algorithms. SLAM is a popular method that can be based on image data or lidar data. The IPIN conference has been an international conference for many years where these topics are discussed. FUSION should also be mentioned. The authors have also described this in their state of the art.

The use of different approaches that can support each other in the field is crucial for meaningful use, as they have different requirements. Visual odometry (Visual SLAM), GNSS-based position estimation and inertial sensors are each very different in terms of their local requirements and stochastic characteristics. This choice of authors for the fusion is therefore optimal in my view. 

It is important that the filter for fusion is provided with the right decision parameters that allow a suitable weighting. In my view, this is the most important added value of such a contribution, in addition to its practical use as a focus. The estimation methods were treated separately here. A future added value could be the implementation in KF or PF.

General information on the paper: The paper does not go into deep detail. 

The overview is largely complete as far as the state of research is concerned and the paper is good and clear. In my opinion, the added value for the reader should be the realization of the fusion. However, it is not possible to implement the system yourself or to work on the content in depth, as the realization of the system is only presented at a conceptual level.

Regarding the achievable accuracy: The comparison to the reference is difficult as a pedestrian. 

An accuracy with cm resolution does not seem realistic to me. I recommend a dm resolution: 1.7 m.  The achievable indoor accuracy of 30cm for the reference is not explained in detail. Where does this value come from? - I think it would have been possible to be much more accurate, but it may not be necessary.

Line 63: The modularity and robustness is given by a fusion algorithm. The selection criteria are decisive as to which approach receives the highest weighting for the position estimate. This can also be: 1 - 0 - 0 ... so that only one approach is used. However, it is more likely to be 0.8 - 0.05 - 0.15, for example 

So you first use a manual selection, presumably also the internal accuracy as a selection criterion, possibly light sensors or similar would also be helpful for an independent assessment, e.g. of the visual odometry.

Figure 9: I like the illustration. However, I would still be interested in the accuracy of the position estimation methods that were not used. It might be possible to assign this to the positions in the graphic or in an additional table below.

Thanks for your paper and good luck for your following in this review process.

Reviewer 2 Report

Comments and Suggestions for Authors

Comments to the Author

This paper proposes a novel approach leveraging three complementary localisation modalities: visual-based, Galileo-based and inertial-based, to maintain responder-safety and optimising operational effectiveness. However, there are several points that need to be addressed to improve the quality of the manuscript.

Suggestions to improve the quality of the paper are provided below:

1.     Beside emergency response, location-based systems have been used in many other application areas including smart energy management, intelligence HVAC controls, point-of-interest identification, and occupancy prediction. Please kindly review and expand upon the following studies on other applications of localisation systems to cater to a more general audience.

Location-based building emergency response

10.1109/IUCC-CSS.2016.013

Location-based smart HVAC controls

https://doi.org/10.1145/2517351.2517370

Location-based occupancy prediction

https://doi.org/10.1016/j.buildenv.2022.109689

2.     When listing out the novelty of the paper, the authors should focus on highlighting the areas where this work extends upon the existing studies instead of describing what is done in this paper.

3.     Beside, visual perception and the utilisation of Galileo signals, fingerprinting-based localisation approaches are also extremely popular in literature, utilising WIFI or Bluetooth Low Energy signals to identify the occupants’ precise location in the building. Please kindly review the following paper as a starting point and mention why this approach is not suitable for this study’s use case: https://doi.org/10.1016/j.buildenv.2020.106681

4.     Since the experiment is conducted in a specific building with a specific experiment path, what is done to ensure that the results from the experiment is representative of the system’s localisation performance.

5.     Please provide a list of limitations of this work and how they will be addressed in future works. Some unanswered questions include:

·      It was mentioned that the visual self-localisation tool has an optional pre-calibration step to get the initialisation parameter. However, it is unclear how the system will perform if the calibration step is not performed. What can be explored in the future to reduce the need for this step?

·      The results in the “Location error at ground control points” figure (missing figure captions) seems to indicate that the system does not perform well in indoor dark due to the higher location error. How can this be addressed in the future?

·      A laptop was used in the experimental setup to host the message broker and had to be carried by the first responder during the experiment. How will this be addressed in future iterations of this system?

Comments on the Quality of English Language

There are no major issues related to the manuscript's quality of English, except for some minor issues highlighted in my current set of comments.

Round 2

Reviewer 2 Report

Comments and Suggestions for Authors

Thank you for taking the time to address my comments thoroughly and comprehensively. I believe all my comments have been adequately addressed, and the quality of the manuscript has increased significantly as a result. I have determined that the manuscript is now ready for publication.

Comments on the Quality of English Language

There are no major issues related to the manuscript's quality of English, except for some minor issues that do not affect the clarity and flow of the manuscript.